# Effectiveness of Statins for Primary Prevention of Cardiovascular Disease in Low- and Medium-Risk Males: A Causal Inference Approach with Observational Data

**DOI:** 10.3390/jpm12050658

**Published:** 2022-04-20

**Authors:** Armando Chaure-Pardos, Isabel Aguilar-Palacio, Mª José Rabanaque, Mª Jesús Lallana, Lina Maldonado, Sara Castel-Feced, Julián Librero, José Antonio Casasnovas, Sara Malo

**Affiliations:** 1Directorate of Public Health, Government of Aragón, 50017 Zaragoza, Spain; achaure@salud.aragon.es; 2Instituto de Investigación Sanitaria de Aragón (IIS Aragón), 50009 Zaragoza, Spain; rabanake@unizar.es (M.J.R.); mjlallana@salud.aragon.es (M.J.L.); lmguaje@unizar.es (L.M.); scastelf@unizar.es (S.C.-F.); jacasas@unizar.es (J.A.C.); smalo@unizar.es (S.M.); 3Grupo de Investigación en Servicios Sanitarios de Aragón (GRISSA), 50009 Zaragoza, Spain; 4Preventive Medicine and Public Health Department, University of Zaragoza, 50009 Zaragoza, Spain; 5Primary Health Care, Servicio Aragonés de Salud, 50017 Zaragoza, Spain; 6Department of Economic Structure, Economic History and Public Economics, University of Zaragoza, 50009 Zaragoza, Spain; 7Unidad de Metodología, Navarrabiomed, Hospital Universitario de Navarra (HUN), Universidad Pública de Navarra (UPNA), IdiSNA, 31008 Pamplona, Spain; julian.librero.lopez@navarra.es; 8Department of Medicine, Psychiatry and Dermatology, University of Zaragoza, 50009 Zaragoza, Spain

**Keywords:** cardiovascular diseases, prevention and control, hydroxymethylglutaryl-CoA reductase inhibitors, comparative effectiveness

## Abstract

In this study, we analyzed the effectiveness of statin therapy for the primary prevention of cardiovascular disease (CVD) in low- and medium-risk patients. Using observational data, we estimated effectiveness by emulating a hypothetical randomized clinical trial comparing statin initiators with statin non-initiators. Two approaches were used to adjust for potential confounding factors: matching and inverse probability weighting in marginal structural models. The estimates of effectiveness were obtained by intention-to-treat and per-protocol analysis. The intention-to-treat analysis revealed an absolute risk reduction of 7.2 (95% confidence interval (CI95%), −6.6–21.0) events per 1000 subjects treated for 5 years in the matched design, and 2.2 (CI95%, −3.9–8.2) in the marginal structural model. The per-protocol analysis revealed an absolute risk reduction of 16.7 (CI95%, −3.0–36) events per 1000 subjects treated for 5 years in the matched design and 5.8 (CI95%, 0.3–11.4) in the marginal structural model. The indication for statin treatment for primary prevention in individuals with low and medium cardiovascular risk appears to be inefficient, but improves with better adherence and in subjectvs with higher risk.

## 1. Introduction

Multiple randomized clinical trials (RCTs) have shown that lipid-lowering statins are effective in reducing cardiovascular disease (CVD) morbidity and mortality in individuals with high risk of CVD [1]. Several secondary analyses of RCT data have shown similar relative efficacy in low- and medium-risk individuals [2,3,4]. However, it is difficult to quantify the effect of statins in absolute terms as this is largely dependent on the characteristics and the baseline risk level of the population studied.

Moreover, RCTs are usually carried out in controlled settings, where the population is carefully selected and subjects are closely followed. These conditions, which are ideal for demonstrating the efficacy of a drug, do not allow us to measure the effectiveness of a drug in real-world conditions. Observational studies that emulate a “target trial” can be used to overcome this limitation, and can provide estimators that are as valid as those of RCTs [5,6,7], with the added advantage of studying a population in the context of real-world clinical practice.

However, non-experimental studies have some limitations. It should be noted that in routine practice, not all individuals with low or medium CVD risk have an indication for statin therapy [8]. Therefore, it is difficult to evaluate the effectiveness of statins in a population with these characteristics using observational data, since certain risk profiles (e.g., very low CVD risk) will be very scarce or entirely absent from the study group. In other words, certain individuals, owing to their baseline characteristics, will have zero or very low probability of receiving the treatment. The absence of a non-zero probability of being assigned to one of the treatment levels indicates that the positivity condition is not fulfilled, and therefore the results obtained may be biased [9]. To avoid this, various statistical techniques such as matching or sample restriction can be used to eliminate subjects with extreme values for key covariates. However, the application of these approaches implies creating different populations, which can lead to distinct outcomes in terms of absolute risk reduction, and can therefore complicate the comparability of the results.

Our objective was to analyze the effectiveness of statin therapy for primary prevention of CVD in low- and medium-risk patients by applying a target trial emulation design and comparing the results of distinct analytical approaches.

## 2. Materials and Methods

We conducted an observational study that emulated the design of several successive clinical trials, as proposed by Hernán et al. [5,10,11], in subjects undergoing treatment between July 2010 and June 2019.

Observational data from the Aragon Workers Health Study (AWHS) cohort [12] were used. The AWHS is a prospective study designed to evaluate the evolution of CVD risk factors and their association with subclinical atherosclerosis in a cohort of 5650 middle-aged workers at an automobile factory in Spain. Follow-up began in 2009 and continues today.

### 2.1. Study Population

In each “trial”, the Systematic Coronary Risk Evaluation (SCORE) [13] was performed for each subject and their cardiovascular risk was calculated according to current European guidelines [8], on the trial start date. After calculating this risk, which combines the SCORE with cholesterol and blood pressure levels, and with history of CVD, diabetes mellitus, and chronic kidney disease, only subjects with low or medium CVD risk, according to the guidelines, were included. Furthermore, in order to only include subjects who were candidates to begin treatment for primary prevention, the following exclusion criteria were applied: (i) subjects who received a statin prescription during the 6 months preceding the trial start date; (ii) subjects with less than 6 months of follow-up; and (iii) subjects who experienced a CVD event at some point prior to the start of the trial. To ensure data quality and to control for confounding factors, subjects for whom there were no data available on tobacco use, body mass index (BMI), systolic blood pressure (SBP), low-density lipoprotein cholesterol (LDL-C), high-density lipoprotein cholesterol (HDL-C), or blood glucose in the previous 12 months were excluded.

### 2.2. Data Sources

Information on pharmaceutical dispensing was obtained from the Aragon pharmaceutical consumption information system. Events were identified using the administrative database of the Minimum Basic Data Set (MBDS), which codes hospital discharges, and the Aragón hospital emergency information system. Data on the number of visits to primary care were obtained from the Aragon primary care information system. The remaining clinical and analytical variables necessary to calculate CVD risk and to control for confusion were obtained from AWHS databases. Mortality data were obtained from the Spanish National Mortality Registry.

### 2.3. Variables Used

The studied drugs were identified using Anatomical Therapeutic Chemical (ATC) codes, as proposed by the World Health Organization in its ATC/DDD Index 2021. ATC codes corresponding to statin therapy were as follows: C10AA (hydroxymethylglutaryl-CoA reductase inhibitors); C10BA (combinations of various lipid modifying agents); and C10BX (lipid modifying agents in combination with other drugs). Statin prescriptions filled on a monthly basis at a pharmacy during the study period were recorded. Patients were considered to have stopped using statins when at least 2 months passed without filling a prescription at a pharmacy.

The first diagnoses of major adverse cardiovascular events (MACE) in the emergency department or upon admission to hospital were considered main events, as well as deaths in which a MACE was the cause of death. To assess the effectiveness of statins in preventing cardiovascular events, a conservative definition of MACE was chosen [14]. Thus, the ICD codes used to identify MACE were I21 and I22 (acute myocardial infarction) for coronary artery disease and I60–I63 (nontraumatic intracranial hemorrhage and cerebral infarction) for CVD.

Information on the following covariates was collected: (i) age at the beginning of the study; (ii) number of visits to primary care in the 6 months prior to MACE; (iii) smoking, divided into 3 categories (smoker, non-smoker, and ex-smoker); (iv) BMI; (v) LDL-C levels; (vi) HDL-C levels; and (vii) blood glucose levels. For each of these covariates each patient was assigned the value recorded prior and closest to the trial start date. LDL cholesterol was calculated using the Friedewald formula [15].

### 2.4. Analyses

Applying the aforementioned selection and exclusion criteria, a clinical trial was emulated for each month between July 2010 and June 2019. The unit of analysis was “subject-trial“, since each subject could participate in more than one trial throughout the follow-up period.

For the selected subjects, two groups were established depending on treatment status during the month the trial began: “initiators” and “non-initiators”. “Initiators” were subjects who began statin treatment during the month the trial began. “Non-initiators” were those who were not receiving statin therapy during the month of study initiation.

Patient follow-up depended on the analysis performed (intention-to-treat or per-protocol analysis). In the intention-to-treat analysis, each patient was followed until the onset of the main event, death, or loss to follow-up, whichever occurred first. In this analysis, each patient remained in the group to which they were assigned at the beginning, regardless of whether they discontinued treatment (in the case of “initiators”) or started it (in the case of “non-initiators”). In the per-protocol analysis, each patient was followed until the onset of the main event, death, loss to follow-up, or deviation from assigned treatment, whichever occurred first. Therefore, in this analysis, “initiators” were censored when they stopped statin treatment and “non-initiators” were censored when they started statin treatment. In all analyses, the first diagnosis of MACE in an emergency episode or upon hospital admission was considered the main event.

To ensure compliance with the positivity condition (i.e., that all subjects had some probability of receiving or not receiving treatment), two distinct approaches were used. The first was matched analysis, whereby each treated subject was matched with an untreated subject from the same trial with similar values for potential confounding variables. This allowed us to obtain effectiveness estimates for a population resembling that which actually receives statin treatment, since in the sample analyzed both treated and untreated subjects have a similar risk of CVD as the treated group. The second approach consisted of sample restriction, whereby subjects with extreme values for key covariates were eliminated. This approach allowed us to obtain effectiveness estimates for the global population with low or medium risk, since the resulting pseudo-population had a risk similar to the global risk of the selected population. To this end, subjects with extreme values for confounding quantitative variables were excluded from the analysis. Extreme values were those that exceeded the maximum value in treated subjects +0.1 standard deviations and those that were below the minimum value in treated subjects −0.1 standard deviations. Using this restricted sample, a marginal structural analysis was performed, creating a pseudo-population by weighting the subjects according to the inverse probability of receiving the assigned treatment (inverse probability weighting). Figure 1 depicts the sample restriction procedure for the variable LDL-C, and shows the distribution of LDL-C for subject-trials assigned to each treatment arm. LDL-C levels are lower in untreated versus treated subjects, and the minimum values correspond exclusively to untreated subjects.

In the population resulting from the matched analysis and in the restricted pseudo-population resulting from the marginal structural model, we calculated the overall incidence, the incidence per treatment group, the difference in incidence, the number of patients needed to treat (NNT) for 5 years to avoid an event, and the incidence ratio, using both intention-to-treat and per-protocol analyses.

All analyses were performed using R version 4.1.1 (2021, The R Foundation for Statistical Computing).

## 3. Results

### 3.1. Intention-to-Treat Analysis

The intention-to-treat analysis included 133,048 subject-trials, corresponding to 4253 subjects. Of these, 473 subject-trials were considered to be treated with statins. Table 1 lists the characteristics of the subject-trials, according to treatment. Appendix A shows the distribution of subject-trials according the type of statin prescribed.

#### 3.1.1. Matched Analysis

The matched analysis included a total of 946 subject-trials (473 pairs). Their characteristics, listed according to treatment group, are shown in Table 2.

In total, 25 events occurred over a total follow-up period of 85,310 months (I = 17.6 events per 1000 subjects followed for 5 years). Among treated subjects, there were 10 events in 42,682 months (I = 14.0 per 1000 subjects followed for 5 years) and among untreated subjects, 15 events occurred in 42,448 months (I = 21.2 per 1000 subjects followed for 5 years).

No statistically significant associations were observed. The absolute risk reduction was 7.2 cases per 1000 subjects treated for 5 years (CI95%, −6.6–21.0 per 1000 subjects followed for 5 years), which implies the need to treat 139 patients for 5 years to avoid a cardiovascular event (5-year NNT = 139, CI95%, −152–48). The incidence ratio of treated to untreated individuals was 0.66 (RR = 0.66; CI95%, 0.30–1.47).

#### 3.1.2. Marginal Structural Model

The pseudo-population created for the marginal structural model included 125,198 subject-trials, of which 441 were considered to be treated. Subject characteristics are shown in Table 3.

In the pseudo-population, 1498.2 events occurred over a total follow-up period of 10,923,864 months (I = 8.23 per 1000 subjects followed for 5 years). In treated subjects, 4.0 events occurred in 39,079.1 months (I = 6.1 per 1000 subjects followed for 5 years) and in untreated subjects, 1494.2 events occurred in 10,884,784.7 months (I = 8.2 per 1000 subjects followed for 5 years).

No statistically significant associations were observed. The absolute risk reduction was 2.2 cases per 1000 subjects treated for 5 years (CI95%, −3.9–8.2 per 1000 subjects followed for 5 years), which implies the need to treat 464 patients for 5 years to avoid a cardiovascular event (5-year NNT = 464; CI95%, −260–123). The incidence ratio of treated to untreated individuals was 0.74 (RR = 0.74, CI95%, 0.28–1.98).

### 3.2. Per-Protocol Analysis

The per-protocol analysis included 133,048 subject-trials, corresponding to 4253 subjects. Of these, 473 subject-trials were considered to be treated with statins.

#### 3.2.1. Matched Analysis

The matched analysis included the same pairs as in the intention-to-treat analysis (473 pairs).

In this analysis, 14 events were recorded in a total of 40,212 months of follow-up (I = 20.89 events per 1000 subjects followed for 5 years). In treated subjects, 1 event occurred in 8014 months (I = 7.5 per 1000 subjects followed for 5 years), and in untreated subjects 13 events occurred in 32,198 months (I = 24.2 per 1000 subjects followed for 5 years).

No statistically significant associations were observed. The absolute risk reduction was 16.7 cases per 1000 subjects treated for 5 years (CI95%, −3.0–36.5 per 1000 subjects followed for 5 years), which implies the need to treat 60 patients for 5 years to avoid a cardiovascular event (5-year NNT = 60; CI95%, −336–27). The incidence ratio of treated to untreated individuals was 0.31 (RR = 0.31; CI95%, 0.04–2.36)

#### 3.2.2. Marginal Structural Model

The pseudo-population created for the marginal structural model included 125,198 subject-trials, of which 440 were considered treated. The characteristics of the subject-trials are shown in Table 3.

In the pseudo-population, 1076.9 events occurred over a total of 9,494,031 months of follow-up (I = 6.81 per 1000 subjects followed for 5 years). In treated subjects, 0.1 events occurred in 7465.1 months (I = 1.0 per 1000 subjects followed for 5 years), and in untreated subjects 1076.8 events in 9,486,566 months (I = 6.8 per 1000 subjects followed for 5 years)

The absolute risk reduction was 5.8 cases per 1000 subjects treated for 5 years (CI95%, 0.3–11.4 per 1000 subjects followed for 5 years), which implies the need to treat 172 patients for 5 years to avoid a cardiovascular event (5-year NNT = 172; CI95%, 3548–88). The incidence ratio of treated to untreated individuals was 0.15 (RR = 0.15; CI95%, <0.01–38.82). Table 4 summarizes the results obtained.

## 4. Discussion

Our results suggest a beneficial effect of statin treatment for primary prevention of CVD in subjects with low or medium risk. Our findings indicate that in order to prevent a cardiovascular event, statins should be prescribed for 5 years to between 139 and 464 patients, depending on the level of risk of the population. Assuming adequate adherence by treated patients, statins should be prescribed to 60–172 patients to prevent a cardiovascular event, depending on the level of risk.

The incidence ratio values estimated using an intention-to-treat analysis are similar to those previously reported in the literature. A meta-analysis carried out by the CCT collaborators using individual data from 27 RCTs [2], reported risk ratios of 0.57–0.77 (depending on the event and risk group) for each reduction in LDL-C of 38.61 mg/dl among subjects with low and medium risk of CVD. Furthermore, Danaei et al. [5], using a study design similar to ours, reported a hazard ratio (HR) of 0.89 between initiators and non-initiators in the general population without discriminating by baseline risk, a result comparable to our IR estimates of 0.66 and 0.74. Compared with previous reports [2,5], the results of our per-protocol analysis appear to overestimate the effect of statins. However, it should be noted that our results may be imprecise given the low number of events included in our analysis, together with the short follow-up of treated subjects (subjects that discontinued treatment were censored), which resulted in very high confidence intervals.

On the other hand, the ARR and NNT results are more difficult to compare, as they depend largely on the specific population analyzed. Glynn et al. [3], in their secondary analysis of an RCT, reported a 5-year NNT of 38, which is far from the value of 60 obtained in our matched per-protocol analysis. While the population included in that study had low levels of LDL-C, both mean age and C-reactive protein levels were higher. This may help explain the higher incidences of CVD in their two treatment groups, and the higher ARR despite similar IR and HR values.

The main limitation of our study is the wide confidence intervals of the results obtained. This is due to the low incidence of cardiovascular events in a population categorized as low- or medium-risk, and the low number of subjects who began treatment during the study period. These differences were exacerbated in the per-protocol analysis due to poor treatment adherence [16,17]. Attempts were made to address these shortcomings by emulating successive trials and using more statistically efficient techniques such as matching. However, our study population of just over 4000 subjects was insufficient to yield more accurate results. Despite this limitation, our estimated incidence ratio values in the intention-to-treat analysis are consistent with those previously reported [2,5], suggesting that the techniques used to avoid confusion proved successful. Therefore, it is reasonable to assume that our estimates of absolute risk reduction in 5 years are also reliable, albeit imprecise.

The marked differences observed between the results of the intention-to-treat and the per-protocol analyses are mainly due to poor statin treatment adherence among treated subjects. In other words, given that the intention-to-treat analysis includes the complete follow-up regardless of adherence and that the per-protocol analysis includes only the follow-up period in which adherence is maintained, the fact that the results are better with “optimal adherence” (i.e., per protocol analysis) than with “suboptimal adherence” (i.e., intention-to-treat analysis) shows the relevance of adherence to the effectiveness of statins. In individuals with low risk of disease, especially in observational studies, poor treatment adherence is expected, since the perceived risk is lower. This poor adherence to statin treatment, which has already been measured in the AWHS population [16,17], results in a treatment persistence of less than 30% at 1 year. By contrast, persistence in RCTs is usually greater than 95% [18]. This should not constitute a problem in the context of the present study, the main objective of which was to evaluate the effectiveness of statin treatment in a real population in real-world conditions. In this sense, our intention-to-treat analysis measures the effectiveness of the medical decision to prescribe, while the per-protocol analysis measures the causal effect of the treatment taken according to the medical prescription.

Another limitation is the possible violation of the positivity condition. To overcome this limitation, we applied two distinct techniques: matching and sample restriction. Because the matched analysis included fewer treated subjects with higher CVD risk, this particular sample had a higher cardiovascular risk than the global population. By contrast, sample restriction yielded a population more similar to the global population, with low and medium cardiovascular risk. Thus, the risk reduction values obtained using the matching approach correspond to the effect obtained with the current prescription system, while those obtained for the pseudo-population (using sample restriction) would better correspond to the effect obtained after treatment of any subject with low or medium CVD risk.

Although our population consisted exclusively of male industrial workers, we believe that our estimators are applicable to the general population, given that our population was selected based on CVD risk profile. Regardless, the inclusion of women, who have a lower risk of CVD, would have rendered the treatment even more ineffective.

Finally, neither of the two approaches resulted in comparable LDL-C levels in the two treatment groups; this parameter was slightly higher in treated subjects. However, if these differences caused confusion, our estimators would underestimate the true effectiveness of statins, since treated subjects would have a higher risk of CVD than untreated subjects. Given that the incidence ratios we obtained were similar to those observed in RCTs [2], such underestimation is unlikely. Additionally, although we know that HBA1c better represents glycemic status, we were forced to use fasting blood glucose as an approximation in our study, since HBA1c was only measured in 30% of blood tests. However, we carried out exploratory analyzes with the subjects who had this information and the results were equivalent to those shown.

Our results suggest that the indication of statin treatment for primary prevention in subjects with low and medium CVD risk is inefficient, given the low adherence observed: all approaches used resulted in high estimated NNT values. The current use of statins in our sample, as represented by the matched analysis, is more efficient than if prescription were extended to all subjects with this cardiovascular profile, as it avoids a greater number of events for the same number of people treated, and can be improved with adequate treatment adherence. It is advisable to take into account these results and those of other similar studies when including specific treatment recommendations in clinical practice guidelines, and to emphasize the need to improve treatment adherence, as this enables more realistic evaluation of the impact of the intervention than in RCTs.

## Figures and Tables

**Figure 1 jpm-12-00658-f001:**
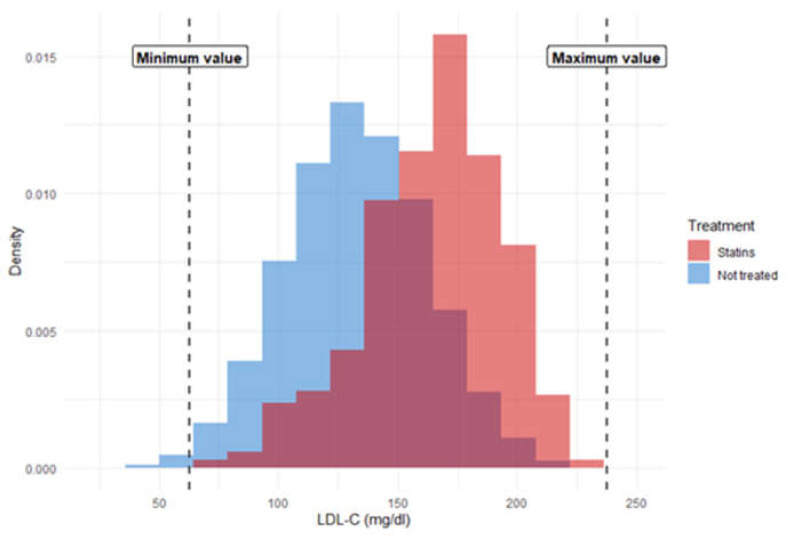
Example of sample restriction for low-density lipoprotein cholesterol.

**Table 1 jpm-12-00658-t001:** Characteristics of the total subject-trials selected for the study.

	Not Treated N = 132,575	Treated with Statins N = 473	*p*
Age, mean (SD)	49.0 (9.1)	52.7 (4.6)	<0.01
Visits to PC in the last 6 months, mean (SD)	0.1 (1.2)	0.5 (2.5)	<0.01
BMI, mean (SD)	27.3 (3.5)	28.1 (3.4)	<0.01
SBP, mean (SD)	123.5 (12.8)	125.3 (13.2)	<0.01
LDL-C, mean (SD)	132.9 (29.4)	164.7 (28.3)	<0.01
Tobacco, *n* (%)			<0.01
Nonsmokers	37,083 (28.0)	105 (22.1)	
Smokers	43,594 (32.9)	155 (32.8)	
Ex-smokers	51,898 (39.1)	213 (45.0)	
HDL-C, mean (SD)	53.2 (11.1)	52.9 (10.7)	0.53
Glucose, mean (SD)	94.5 (10.7)	96.1 (11.7)	<0.01
Follow-up time ^a^, mean (SD)	87.5 (19.0)	90.6 (18.2)	<0.01

N, total number of subjects per treatment group; *p*, *p*-value of the Student’s *t*-test or Chi-squared test for the variables Smokers and Ex-smokers; SD, standard deviation; *n*, number of subjects per category; PC, primary care; BMI, body mass index: SBP, systolic blood pressure; LDL-C, low-density lipoprotein cholesterol; HDL-C, high-density lipoprotein cholesterol. ^a^ Months of follow-up in the intention-to-treat analysis.

**Table 2 jpm-12-00658-t002:** Characteristics of the total subject-trials in the matched analysis.

	Not Treated N = 473	Treated with Statins N = 473	*p*
Age, mean (SD)	52.6 (4.6)	52.7 (4.6)	0.66
Visits to PC in the last 6 months, mean (SD)	0.4 (2.1)	0.5 (2.5)	0.51
BMI, mean (SD)	28.0 (3.1)	28.1 (3.4)	0.82
SBP, mean (SD)	125.2 (12.0)	125.3 (13.2)	0.94
LDL-C, mean (SD)	159.6 (25.7)	164.7 (28.3)	0.04
Tobacco, *n* (%)			0.96
Nonsmokers	107 (22.6)	105 (22.1)	
Smokers	151 (31.9)	155 (32.8)	
Ex-smokers	215 (45.5)	213 (45.0)	
HDL-C, mean (SD)	53.0 (9.4)	52.9 (10.7)	0.88
Glucose, mean (SD)	95.8 (10.7)	96.1 (11.7)	0.67
Follow-up time ^a^, mean (SD)	89.7 (19.2)	90.6 (18.2)	0.47

N, total number of subjects per treatment group; *p*, *p*-value of the Student’s *t*-test or Chi-squared test for the variables Smokers and Ex-smokers; SD, standard deviation; *n*, number of subjects per category; PC, primary care; BMI, body mass index: SBP, systolic blood pressure; LDL-C, low-density lipoprotein cholesterol; HDL-C, high-density lipoprotein cholesterol. ^a^ Months of follow-up in the intention-to-treat analysis.

**Table 3 jpm-12-00658-t003:** Characteristics of the total subject-trials in the pseudo-population.

	Not Treated N = 124,757	Treated with Statins N = 441	*p*
Age, mean (SD)	50.1 (7.8)	51.8 (5.9)	<0.01
Visits to PC in the last 6 months, mean (SD)	0.1 (1.1)	0.2 (1.4)	0.79
BMI, mean (SD)	27.5 (3.4)	27.7 (3.3)	0.20
SBP, mean (SD)	123.7 (12.6)	125.1 (13.6)	0.03
LDL-C, mean (SD)	134.9 (28.3)	140.7 (27.8)	<0.01
Tobacco, *n* (%)			0.93
Nonsmokers	34,920 (28.0)	121 (27.3)	
Smokers	39,197 (31.4)	137 (31.1)	
Ex-smokers	50,640 (40.6)	183 (41.5)	
HDL-C, mean (SD)	53.2 (10.9)	53.2 (11.3)	0.99
Glucose, mean (SD)	94.8 (10.8)	95.1 (11.8)	0.69
Follow-up time ^a^, mean (SD)	87.2 (19.1)	88.6 (18.6)	0.10
Follow-up time ^b^, mean (SD)	76.04 (28.96)	16.93 (27.43)	<0.01

N, total number of subjects per treatment group; *p*, *p*-value of the Student’s *t*-test or Chi-squared test for the variables Smokers and Ex-smokers; SD, standard deviation; *n*, number of subjects per category; PC, primary care; BMI, body mass index: SBP, systolic blood pressure; LDL-C, low-density lipoprotein cholesterol; HDL-C, high-density lipoprotein cholesterol. ^a^ Months of follow-up in the intention-to-treat analysis. ^b^ Months of follow-up in the per-protocol analysis.

**Table 4 jpm-12-00658-t004:** Summary of the results.

Follow-Up	Type of Analysis	ARR (5 Years × 1000 People)	NNT	IR
Intention-to-treat	Matched	7.2 (−6.6–21.0)	139 (−15–48)	0.66 (0.30–1.47)
	Marginal structural	2.2 (−3.9–8.2)	464 (−26–123)	0.74 (0.28–1.98)
Per protocol	Matched	16.7 (−3.0–36.5)	60 (−336–27)	0.31 (0.04–2.36)
	Marginal structural	5.8 (0.3–11.4)	172 (3548–88)	0.15 (<0.01–38.82)

ARR, absolute risk reduction; NNT, number of patients needed to treat for 5 years to avoid an event; IR, incidence ratio.

## Data Availability

Data available under request.

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
