# Peer review of "Effectiveness of Statins for Primary Prevention of Cardiovascular Disease in Low- and Medium-Risk Males: A Causal Inference Approach with Observational Data"

_jpm, 2022, doi:10.3390/jpm12050658_

Round 1
Reviewer 1 Report
An interesting study which emphasize the importance of chosen statistical method on data interpretation and, obviously, on clinical repercussions.
In my opinion the title should be modified. The "workers" word has no value or significance to the reader. It seems that "male" descriptor may be more relevant.
In the 2.1 "study population" section the authors describe that a risk score has been calculated using the CVD history as one of the factors, but those who experienced a CVD event prior to trial start have been excluded. It seems to me that the concept should be reanalyzed.
Glucose as a covariate in the analysis may be problematic. The value of this parameter may not be representative in the given context, while HBA1C could underscore better the glycemic status.
One of the factors discussed as having mayor impact on the result was treatment adherence to statins. I believe that such data has to be presented for the different models described.
Author Response
Please, see the attachment.

Reviewer 2 Report
Dear authors,
I read carefully the entire manuscript. It is a nice and sustained work. But I have some comments on it.
Why You define MACE just as myocardial infarction and nontraumatic intracranial haemorrhage and cerebral infarction as is stated in the manuscript? In majority of all published studies and data available, MACE is a composite term which include total death, myocardial infarction, coronary revascularization, unstable angina requiring hospitalization, stroke, and hospitalization related to the heart failure. It looks like there where not other cardiovascular events in the entire population studied during follow-up period.
Another problem for that I need more details: during the study no any studied patient and included in the initial work-up developed diabetes mellitus?
Can You present some data regarding individual response of LDL-cholesterol during statin therapy period?
"Statins"is a quite general term. Can you provide more informations regarding the type and dose of statin that were used? The response to a specific dose of a specific statin was optimal or suboptimal? Were there any fluctuations in the type of statins or dosage during the study?
Regards,
Reviewer
Author Response
Please, see the attachment.

Round 2
Reviewer 1 Report
The authors made adequate changes.